# The Spherical Inverted Pendulum: Exact Solutions of Gait and Foot Placement Estimation Based on Symbolic Computation

Giuseppe Menga 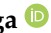

Department of Control and Computer Engineering, Politecnico di Torino, Corso Duca degli Abruzzi 24, 10129 Torino, Italy; menga@polito.it; Tel.: +39-011-090-7272

**Abstract:** The gait and the Foot Placement Estimation (FPE) has recently been extended to 3-D spaces by adopting a specific form of a spherical inverted pendulum (SIP). The approach is very attractive, as it does not involve dynamics, but it is based solely on energies and momenta, however the authors (DeHart et al.) introduced several questionable approximations, in order to reach a manageable solution. The scope of the present paper is to revisit this spherical inverted pendulum applied to biped walking, offering an exact solution to the gait and the FPE by using symbolic computation. This is facilitated by exploiting the Kane's approach to dynamical modelling, and his software environment for symbolic manipulation, called Autolev. It generates explicit formulas describing the energies and angular momenta before/after the impact, along with the mechanics of the impact. As the resulting equations, function of (measurable) angular positions and velocities, are very compact, embedded in a numerical nonlinear solver, are suitable to be implemented in real time and used in practice to control biped robots or lower limb exoskeletons. The two main contributions of the paper are: the recovery of the balance by stepping, in the presence of a push in an arbitrary direction and omnidirectional walking. In this last respect, this specific form of SIP emphasizes the expenditure of energy in the walk. For the first time, at our knowledge, the walk of the SIP, based on energy, has been compared to the simulation of a 12 degrees of freedom biped robot tracking preview signals using the Zero Moment Point (ZMP) of the Linear Inverted Pendulum (LIPM). This quantitatively shows the inefficiency, in terms of energy, of the ZMP-based walk, and the gain due to the recovery of the collision of the flying foot. Similarity in the sagittal plane and differences in the frontal plane of the center of mass trajectories of the two approaches are shown, to open the road to an integration of fully actuated and underactuated controls, for an efficient full-dimensional robot gait to be developed in a future paper.

**Keywords:** humanoid and bipedal locomotion; legged robots; passive walking; foot placement estimation

## 1. Introduction

### 1.1. Background

In the last decade, especially driven by the robotic school of the University of Waterloo, the inverted pendulum model, as an evolution of the passive pantograph walker, has been proposed for foot placement estimation (FPE) and the related biped gait design. First, the problem was solved in two dimensions [1,2], with a first extension and test in 3-D in [3]. More recently, the approach has been transferred to 3-D adopting a specific form of a spherical inverted pendulum (SIP) and called SFPE [4,5]. However, in order to make the problem manageable several questionable simplifications were introduced, i.e. the projections of the central inertia on the two rotation axes of the SIP are considered constant, the rotation velocity does not change before and after the contact of the swing foot with the ground, the impact is approached approximatively, so it is the computation of the angles after the impact.

The solution of the problem depends on three phases: pre-impact, impact and post-impact of the flying foot. The total energy and the angular moment on the pivot point

projected on the vertical axis are constant in the periods before and after the impact, the velocities and loss of energy after the impact can be calculated, finally, the equilibrium is reached imposing zero to both angular velocities at the erect standing balance point.

Instead of introducing approximations in these three steps, an exact solution is possible by using explicitly the mathematical equations of all involved variables, and processing them with a numerical solver, to find a solution.

### 1.2. Symbolic Computation

The programming of the exact expressions of the previously mentioned three steps is simplified by adopting a method to describe dynamical systems introduced at the end of the last century by Prof. Kane of Stanford, known as the Kane's method [6]. He also developed a symbolic manipulation software environment, called Autolev (now MotionGenesis) [7], to support his method and to generate fragments of very efficient code of all needed mathematical expressions to be embedded into a nonlinear numerical solver. The approach allows to represent unitarily, either holonomic and non-holonomic systems, and to handle explicitly kinetic energy, momenta, impact, impulsive forces, and generalized momenta.

### 1.3. State of the Art of Balance, Stepping and Walking

In humanoid robotics the generality of the techniques covers basic walking (flat-footed) on flat surfaces in the absence of disturbances. They mostly track for the whole stride a preview signal based on the ZMP of the LIPM [8–10].

At difference, human-like gait, with its mix of fully actuated and underactuated phases (where walking during one of the phases is a "controlled falling") is more complex [11].

Push recovery, walking on rough terrain, and agile footstep control are active research topics [12].

For push recovery in 2-D, along with [2], using the LIPM, see also [13], and, by adding a flywheel [14]. In 3-D the concept of "N step-capturabiliy" has been introduced in [15]. However, the LIPM is used, i.e., the hight of the COG is assumed constant, and no inertia is accounted for.

Ref. [11] discusses theoretically the problems of underactuation and collision in the walk, applied in practice in this paper. The very recent reference [12] contains a comprehensive review of the literature in this field and approaches the walk, as is done here, through foot placement.

### 1.4. Aims and Organization of the Paper

The novel approach of processing with a numerical solver the basic energy equations of the SFPE does not involve dynamics and offers several advantages in generating the walk. It does not follow an a priory trajectory, the gait style can be changed at each step, so also cadence and step length, allowing aperiodic walk. It is robust to disturbances. Flat ground has been considered here for simplicity; however, looking one step ahead (watch your step!), also non-flat ground can be accounted for. Maneuvering has been demonstrated in the last section.

This paper has not the intention to offer a solution for the control of a complete robot, but to pave the road, with a motion generation, for a future integration in the gait of three aspects: a realist stride with finite double stance periods, energy efficiency, and a mixture of fully actuated and underactuated phases. The idea to add a small energy at each step to maintain the walk is similar to passive-dynamic walkers [16], and in the line of the frameworks of hybrid zero dynamics [17–19].

The paper is organized as follows: Section 2 discusses the existing results of the SFPE; Section 3 introduces the Kane's method and the symbolic environment Autolev; Section 4 presents the spherical inverted pendulum (SIP) model used in this paper; Section 5 describes the equations needed to estimate the foot placement to reach the balance point; Section 6 applies the approach to find the balance point by stepping in the presence of disturbances in any direction; Section 7 applies the SIP to generate a gait with arbitrary

trajectory and pace, and compares the gait generated from the SIP with the classical gait of an equivalent biped robot based on the ZMP of the LIPM and previews signals [20]; Section 8 concludes the paper, and outlines future works to embed the SIP in the generation of the gait for a complete biped.

## 2. The Spherical Foot Placement Estimation

Traditionally, the two degrees of freedom of the SIP (sometime three degrees of freedom are defined, involving, also, the length of the pendulum) were obtained with two rotations on the horizontal axes of the inertial frame [8]. As the final objectives were the two projections of the COG trajectory on the ground, this model originated the celebrated ZMP expressions of the LIPM [9].

In the present case, in order to exploit energies and momenta, rotations along the vertical and one of the horizontal axes have been chosen. Indicated with $\gamma$ and $\omega$ the rotation velocities of the SIP, on the vertical and the horizontal axis on the frontal plane of the biped, the approach of [4,5] is based on the projection of the angular momentum on the pivot point of the pendulum on these two directions, and expressing the kinetic energy as function of these two projections. Cleverly, noting that the total energy and the momentum projected on the vertical axis remain constant during the periods before and after the collision of the swing foot with the ground, approaching the collision and the switching of the pivot foot, the problem is solved by writing the equations of the pre-impact, impact and post-impact phases.

### Discussion

To render the problem manageable the two projections of the angular momentum are expressed as a function exclusively of $\gamma$ or $\omega$ and the two projections of the central inertia are considered constants. i.e., the central inertial matrix is assumed diagonal, and $I_{11}$ and $I_{33}$ have identical values (see Appendix A). The impact is solved approximatively. Moreover, no importance is given to the angle of rotation on the vertical axis.

## 3. The Kane's Method and Autolev

In this work, the so-called Kane's method [6] was adopted to model the spherical inverted pendulum. This method is particularly interesting in this case because it is equally applicable to either holonomic and non-holonomic systems and, for non-holonomic systems, without the need to introduce Lagrangian multipliers. Briefly, the main contribution of the Kane's method is that, through the concepts of motion variables (later called generalized speeds), the vectors of partial velocities and partial angular velocities, generalized active forces and generalized inertia forces, the dynamical equations are automatically determined, enabling forces and torques with no influence on the dynamics to be eliminated early in the analysis. Early elimination of these noncontributing forces and torques greatly simplifies the mathematics and enables problems with greater complexity to be handled.

### 3.1. Generalized Coordinates and Speeds

A multi-body system, which possesses $n$ degrees of freedom, is represented by a state with a $n$-dimensional vector **q** of configuration variables (*generalized coordinates*) and an identical dimension vector **u** of *generalized speeds* called also *motion variables*, that could be any nonsingular combination of the time derivatives of the generalized coordinates that describe the configuration of a system. These are the kinematical differential equations:

$$u_r = \sum_{i=1,\cdots,n} Y_{ri}\dot{q}_i, r = 1, \cdots, n \qquad (1)$$

$Y_{ri}$ may be in general nonlinear in the configuration variables so that the equations of motion can take on a particularly compact (and thus computationally efficient) form with the effective use of generalized speeds.

### 3.2. Partial Velocities and Angular Velocities

Partial velocities of each point (partial angular velocity of each body) are the $n$ three-dimensional vectors expressing the velocities of that point (angular velocity of that body) as a linear combination of the generalized speeds. Let be $\mathbf{v}^B$ the translational velocity of a point $B$ and $\boldsymbol{\omega}^P$ the rotational velocity of a body $P$ with respect to the inertial reference frame, then

$$\begin{aligned} \mathbf{v}^B &= \sum_{r=1\ldots n} \mathbf{v}_r^B u_r \\ \boldsymbol{\omega}^P &= \sum_{r=1\ldots n} \boldsymbol{\omega}_r^P u_r \end{aligned} \tag{2}$$

where $\mathbf{v}_r^B$ and $\boldsymbol{\omega}_r^P$ are the $r$th partial velocity and partial angular velocity of $B$ and $P$, respectively.

### 3.3. Generalized Active and Inertia Forces

The $n$ generalized forces acting on a system are constructed by the scalar product (projection) of all contributing forces and torques on the partial velocities and partial angular velocities of the points and bodies they are applied to.

Let us consider a system composed by $N$ bodies $P_i$, where the torque $\mathbf{T}^{P_i}$, and force $\mathbf{R}^{B_i}$ applied to a point $B_i$ of $P_i$ are the equivalent resultant ("*replacement*" [6]) of all active forces and torques applied to $P_i$. Then

$$F_r^{P_i} = \boldsymbol{\omega}_r^{P_i} \cdot \mathbf{T}_i^{P_i} + \mathbf{v}_r^{B_i} \cdot \mathbf{R}_r^{B_i} \tag{3}$$

is the $r$th generalized active force acting on $P_i$ and

$$F_r = \sum_{i=1,\cdots,N} F_r^{P_i} \tag{4}$$

the $r$th generalized active force acting on the whole system. Identically for the inertia forces, indicated as $F_r^*$.

The dynamical equations for an $n$ degree of freedom system are formed out from generalized active and inertial forces $F_r^*$

$$F_r + F_r^* = 0, r = 1, \cdots, n. \tag{5}$$

These are known as Kane's dynamical equations.

They result in a $n$-dimensional system of second order differential equations ($2n$ order state variable representation) on generalized coordinates and speeds

$$\bar{\mathbf{M}}(\mathbf{q})\dot{\mathbf{u}} + \bar{\mathbf{C}}(\mathbf{q}, \mathbf{u})\mathbf{u} + \bar{\mathbf{G}}(\mathbf{q}) - \bar{\boldsymbol{\Gamma}}(\mathbf{q}, \mathbf{u}, \boldsymbol{\tau}) = \mathbf{0}, \tag{6}$$

where the parameter definitions are similar but not identical of the classical Lagrangian form and more efficient computationally [21].

### 3.4. Non-Holonomic Constraints

When $m$ constraints on the motion variables are added to the model, only $n - m$ generalized speeds are independents. The system is, then, called a non-holonomic system. The *non-holonomic* constraints are expressed as a set of $m$ linear relationships between dependent and independent generalized speeds of the type

$$u_r = \sum_{i=1,\cdots,p} A_{ri} u_i, r = p+1, \cdots, n, \tag{7}$$

with $p = n - m$. In this case, selected the independent speeds, the Kane's method immediately offers the minimal $2p$ order state variable representation from

$$\widetilde{F}_r + \widetilde{F}_r^* = 0, r = 1, \cdots, p, \tag{8}$$

where Kane calls $\widetilde{F}_r$ and $\widetilde{F}_r^*$ non-holonomic generalized active and inertial forces, while the remaining $m$ original redundant equations resolve themselves in the expressions of the $m$ reaction forces/torques returned by the constraints. Because the Kane's method is fundamentally based on the projection of forces on a tangent space on which the system dynamics are constrained to evolve, spanned by the partial velocities, reaction forces/torques result from the projection on its null-space.

Moreover, it is always possible to handle an holonomic (configuration) constraint as if it is non-holonomic, that is, to treat it as a motion constraint. This is particularly advantageous to represents the spherical inverted pendulum with a pantograph during a step, where in the first phase non-holonomic constrains allow pivoting on the supporting leg, and in the second phase, releasing the non-holonomic constraints the impact of the swing leg with the ground can be represented.

### 3.5. Unilateral Constraints and Collision

As a consequence of switching between different non-holonomic models during gait, unilateral constraints and collisions cannot be ignored.

Clearly, adopting non-holonomic dynamics assuming points of the feet fixed to the ground is valid for bilateral constraints (ignoring eventual detachment from the ground and slipping). In the approaches known as hybrid complementarity dynamical systems based on forward dynamics [22] the necessary conditions for satisfying unilateral constraints are directly embedded into the model. Vice versa, a minimalistic view is adopted here, noting that in a physiological gait, normally, bilateral constraints on the feet are not assumed to be violated. Hence, we design a priori walking strategies and we test through the simulator that this effectively occurs, by monitoring, a posteriori, reaction forces for the conditions:

$$F_{z_{foot_i}} > 0, i = 1, 2 \tag{9}$$

and

$$|F_{j_{foot_i}}| < \mu F_{z_{foot_i}}, j = x, y, i = 1, 2. \tag{10}$$

Obviously, the control we propose cannot adapt itself to pathological conditions, such as a slipping surface.

For the second point, mechanics of the collision of the swing foot to the ground has to be considered, when switching to the next step causes the transfer of final conditions of the generalized speeds of one phase to the initial conditions of the successive. With reasonable assumptions of non-slipping and anelastic restitution the reaction impulsive force $\mathbf{F}^B$ at the impact point $B$ and the initial conditions of the generalized speeds for the new phase $\mathbf{u}(t^+)$ can be computed. Also for this aspect, Autolev offers all needed mechanical expressions.

The following analysis is based on two concepts: *generalized impulse* and *generalized momentum* [6,23]. Indicate, as usual, with $\mathbf{v}_r^B$ the r-th component of the partial velocity vectors of the point $B$ (the swing foot), the *generalized impulse* at the point $B$ at the contact with the ground at instant $t^-$ is defined as the scalar product of the integral of the reaction impulsive force $\mathbf{F}^B \delta(t - \tau)$ in the time interval $t^- \div t^+$ with the corresponding partial velocities

$$I_r \approx \mathbf{v}_r^B(t^-)^T \cdot \mathbf{F}^B, r = 1, \cdots, n, \tag{11}$$

the *generalized momentum* is defined as the partial derivative of the kinetic energy $K$ with respect to the r-th generalized speed

$$p_r(t) = \partial K / \partial u_r, r = 1, \cdots, n, \tag{12}$$

then, Kane proves that

$$I_r \approx p_r(t^+) - p_r(t^-). \tag{13}$$

Indicate the matrices

$$\mathbf{V}^B = (\mathbf{v}_1^B(t^-) \cdots \mathbf{v}_n^B(t^-)) \tag{14}$$

$$\mathbf{P} = \{\partial p_i(t^-)/\partial u_j\}, i, j = 1, \cdots, n \tag{15}$$

of vectors of partial velocities, and of partial derivatives of $p_r(t)$ with respect to the generalized speeds, and the vectors

$$\mathbf{I} = [I_1 \cdots I_n]^T = \mathbf{V}^{B^T} \cdot \mathbf{F}^B \tag{16}$$

$$\mathbf{u}(t) = [u_1(t) \cdots u_n(t)]^T \tag{17}$$

$$\mathbf{v}^B(t) = \mathbf{V}^B \cdot \mathbf{u}(t) \tag{18}$$

$$[p_1(t), \cdots, p_n(t)]^T = \mathbf{P} \cdot \mathbf{u}(t) \tag{19}$$

of *generalized impulses*, of *generalized speeds*, of the velocity of point $B$ and of *generalized momenta*, respectively.

Then, taking into account from (16) to (19), considering that $\mathbf{v}^B(t^-)$ is known and $\mathbf{v}^B(t^+)$ is zero, assuming non-slipping condition and inelastic collision, the following system of equations is solved to derive the unknown $\mathbf{F}^B$ and $\mathbf{u}(t^+)$:

$$\begin{bmatrix} -\mathbf{P}\mathbf{u}(t^-) \\ 0 \end{bmatrix} = \begin{bmatrix} \mathbf{V}^B(t)^T & -\mathbf{P} \\ 0 & \mathbf{V}^B(t) \end{bmatrix} \cdot \begin{bmatrix} \mathbf{F}^B \\ \mathbf{u}(t^+) \end{bmatrix} \tag{20}$$

An essentially similar equation was discussed in [11]. At the solution, along with the velocity $\mathbf{u}(t^+)$ after the impact, it must be verified that the impulsive force $\mathbf{F}^B$ satisfies the conditions of unilateral constraint (9) and (10).

## 4. The Spherical Inverted Pendulum Model

In describing the spherical inverted pendulum the same notation used in [4,5] was adopted. In addition, explicitly, $\theta_z$ indicates the angle of rotation with respect to the vertical axis, and two degrees of freedom of the swing leg relative to the pendulum were introduced, with the angles $\alpha_z$ and $\alpha$, as shown in Figure 1, with the kinematics of the joints in Figure 2a. The configuration Figure 2b will be used in special situations, only to perform side shuffle.

The angle position and velocity of the pendulum on the z and y axes of the inertial frame, and the two rotations of the swing leg with respect to the local axis z, and y of the supporting leg are $\theta_z, \theta, \gamma, \omega, \alpha_z, \alpha$, respectively. The configuration variables of the model are $\theta_z, \theta, x, y, z$, as the swing leg is considered without mass and inertia, where $x, y, z$ are the coordinates of the pivot foot. The motion variables are $\gamma, \omega, u_1, u_2, u_3$, where imposing a non-holonomic constraint to the pivot foot, the velocities, $u_1 = \dot{x}, u_2 = \dot{y}, u_3 = \dot{z}$, are zero during the swing phase, but are released at the impact of the swing foot with the ground.

In the next sections the SFPE and the gait based on this model are described.

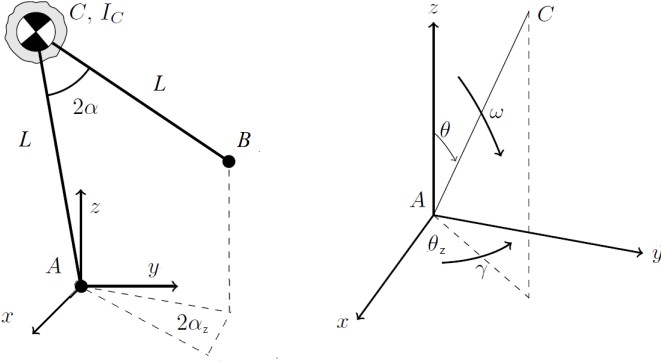

**Figure 1.** The spherical inverted pendulum.

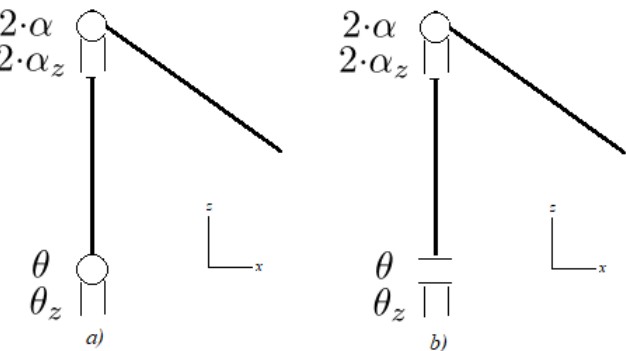

**Figure 2.** The kinematics of the joints—(**a**) forward/bacward motion of the flying leg, (**b**) side shuffle.

### 5. The Estimation of the Balance Point

Before the impact the motion variables have value $\gamma^-, \omega^-, 0, 0, 0$ and after $\gamma^+$, $\omega^+, u_1^+, u_2^+, u_3^+$. The total energy and the projection on the vertical axis of the angular momentum, $k^\gamma$ (the notation of [4] is maintained, even if it is shown in Appendix A that in general both speeds are present in this projection, and here an explicit dependency on $\gamma$ is no more needed), are constant before and after the impact, however, they have a reduction during the impact.

To simplify the computations and to avoid spurious solutions, the next procedure is started after the pendulum reaches the vertical position and $\theta > 0$. Let us say that at time $t_0$ the state variables assume the values $\theta_{z0}, \gamma_0, \theta_0, \omega_0$, the total energy $T_0$, and the moment on the vertical axis $k^\gamma{}_0$ (these last two values are the same, also, at the unknown instant of the impact $t^-$). This gives the first equation, linking all state variables at the pre-impact.

$$T_0 = T(\theta_z{}^-, \gamma^-, \theta^-, \omega^-) \tag{21}$$

At the impact the swing foot, indicated with the point B, touches the ground. The vertical coordinate of $B$ offers the second equation, linking the pre-impact angle $\theta^-$ to $\alpha$ and $\alpha_z$

$$B_z(\theta^-, \alpha, \alpha_z) = 0 \tag{22}$$

The constant momentum $k^\gamma$ offers the third equation, linking $\gamma^-$ to the other pre-impact motion variable

$$k^\gamma{}_0 = k^\gamma(\gamma^-, \theta^-, \omega^-) \tag{23}$$

Switching the pivot foot after the impact, the relationship between the angles $\theta_z{}^+, \theta^+$ of the new pivot leg from $\theta_z{}^-, \theta^-, \alpha, \alpha_z$ is obtained equating the three projections, with respect to the inertial axes, of swing and support legs (i.e., the swing leg becomes the new support leg)

$$SU(\theta_z{}^+, \theta^+) = SW(\theta_z{}^-, \theta^-, \alpha, \alpha_z) \tag{24}$$

The solution of the impact equation (performed symbolically) (20) gives the motion variables after the impact, hence the total energy and the angular momentum $k^\gamma$. The total energy after the impact can be evaluated before the switching of the pivot foot so it does not require the value of $\theta^+$ and $\theta_z^+$ after switching. The angular momentum is computed on the new pivot point, so it requires the new value of $\theta^+$ and $\theta_z^+$ after the switching. This gives the third and fourth equations.

$$TE^+ = TE(\theta_z{}^-, \theta^-, \gamma^+, \omega^+, u_1+, u_2+, u_3+)$$
$$k^{\gamma+} = k^\gamma(\theta_z{}^+, \gamma^+, \theta^+, \omega^+) \tag{25}$$

Moreover, by imposing velocity zero of the swing foot, after the impact, angles before the impact can be related to motion variables after, with a further relationship

$$[\dot{B}_x, \dot{B}_y, \dot{B}_z]^T = 0 = F(\theta_z{}^-, \theta^-, \alpha, \alpha_z, \gamma^+, \omega^+, u_1{}^+, u_2{}^+, u_3{}^+) \tag{26}$$

To estimate the foot placement to reach the balance in an erect posture after the impact, with $\omega = 0, \gamma = 0$, and $\theta = 0$, noting that $k^{\gamma+}$ is zero by the last condition (28), it is imposed that the total energy after the impact is equal to the maximal potential energy

$$TE^+ = m \cdot g \cdot L \tag{27}$$

Finally, to impose that $\gamma$ be zero at the balance point (but not necessarily after the impact, as it will be seen in the next Figure 4), from the impact the last equation is set

$$k^{\gamma+}(\theta_z{}^+, \gamma^+, \theta^+, \omega^+) = 0 \tag{28}$$

From the previous relationships, the unknown variables $\theta_z{}^-, \theta_z{}^+, \gamma^-, \theta^-, \theta^+, \omega^-, \alpha, \alpha_z$ are determined, using non-linear least squares, with some numerical solver such as the Levenberg-Marquardt algorithm [24,25].

## 6. Recovering Balance

The first application of the SFPE is to recover balance in the presence of an impulsive disturbance. It is assumed that the biped is in quiet standing balance, conventionally oriented in the direction of the x axis. An impulsive horizontal force in an arbitrary direction generates a velocity of the COG. The biped, in reorienting himself, will react in two different ways, with a minimal rotation around the z axis, according to the relative direction of the pulse: if the direction is closer to his sagittal plane he will mostly move the free leg forward (or backward), if it closer to the frontal plane the motion will be mostly laterally with a side shuffle. To emulate these two distinct situations if the direction is less than 45° to the sagittal plane the model adopted is the standard one of Figure 2a, otherwise Figure 2b. When the angle $\theta$ falls below a safety value and the direction is detected from the falling velocity, let say $[v_x, v_y, v_z]^T$, the model (a) or (b) is selected according to $atan2(v_y, v_x)$, and initial velocities are assigned to $\omega$ and $\gamma$ from the inverse of the first two rows of the matrix of partial velocities (29)

$$\mathbf{V}^{COG} = [\mathbf{v}_\omega^{COG} \mathbf{v}_\gamma^{COG}]. \tag{29}$$

Then, the SFPE algorithm is run. This defines, among other variables, the swing foot angles $\alpha, \alpha_z$ for the impact that allows recovering the balance in one step. The numerical algorithm does not converge in two cases: when the total energy after the impact is lower than the maximum potential energy, or when one step is not enough to recover the balance. In the first case the condition (27) cannot be satisfied. In the second case the bound on the maximum allowable $\alpha$ is not satisfied.

Two examples are presented using the two models (Figure 3). The situation of a push perfectly aligned with either the sagittal or the frontal plane is not considered, as it can be simply solved with the classical 2-D approach. In both cases the initial velocity is of 0.5 m/s and it is detected, at instant 0.5 s, when the falling angle reaches 0.1 rad. The first example, using model (a), presents the response to a push at an angle of 20° from the x axis, in the second, with model (b), the angle is of 110°.

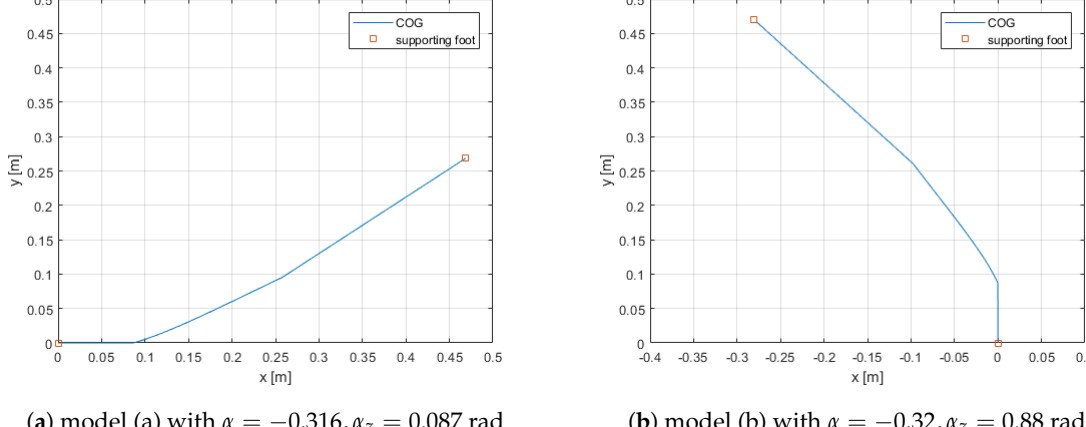

(**a**) model (a) with $\alpha = -0.316, \alpha_z = 0.087$ rad    (**b**) model (b) with $\alpha = -0.32, \alpha_z = 0.88$ rad

**Figure 3.** Recovering an impulsive disturbance-COG behaviours.

Apart from a difference in the sign of the angles, and the velocities the two behaviours are very similar, so the case of model (b) is detailed, only.

The original central inertia matrix of the example, the same of the biped of [20], given in the Appendix A, had the elements I12 and I23 equal zero, the correct responses of angles (speeds and positions) in Figure 4a,b are represented with solid lines. To see the differences, fictitious values different from zero have been assigned to I12 and I23, and the example rerun. The responses are indicated in the legends with a X, and plotted with dashed lines.

From the SFPE, the flying foot final position can be forecasted and the SIP COG trajectory used as a reference, knowing the kinematics, to control the joints of a real biped robot or exoskeleton to recover balance.

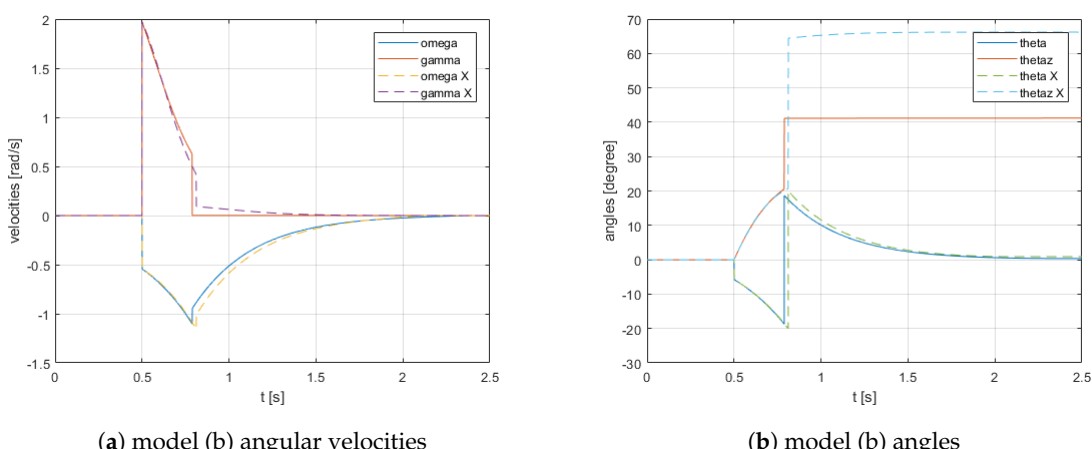

(**a**) model (b) angular velocities    (**b**) model (b) angles

**Figure 4.** Angle velocity and position behaviours-with the effect of off-diagonal elements different from zero in the inertial matrix.

## 7. The Gait

At difference of other works, only some of the expressions of Section 5 are exploited for generating the gait, the SFPE is only used to impose a halt at the end of the walk.

$\alpha$ is chosen to achieve the desired step length, $\alpha_z$ to achieve the COG sway from $-\theta_{zMax}$ to $\theta_{zMax}$ and to control the offset with respect to the baseline of walk. The gait is initiated giving an initial condition to $\theta_z, \gamma, \theta, \omega$, or, simply, from a standing up balance by leaving the pendulum to fall forward.

Each step is concluded when the swing foot touches the ground (the vertical coordinate of point B becomes zero, Equation (22)). From the impact Equation (20) the new motion variables $\gamma^+, \omega^+, u_1^+, u_2^+, u_3^+$ are determined, and from Equation (24) the starting values

of $\theta_z{}^+, \theta^+$ for a new step are computed. The gait is maintained by increasing at each step, after impact, the resulting $\omega^+$ to compensate for the reduction of kinetic energy due to the impact, controlling, also, the gait cadence, and perturbing $\gamma^+$ to correct the angle of direction of the walk.

In the present model, the two legs have no mass and inertia. Therefore, the motions of the angles $\alpha$ and $\alpha_z$ are instantaneous and energy free. The only energy contribution to maintain the walk is given by proper impulsive forces and torques just after the impact to modify the velocities $\omega^+$ and $\gamma^+$ resulting from the impact. This emulates, in a real walk, the contribution given by the biped in the brief double support phase and in the period of single support when the foot is flat and able to transfer torques.

Five control variables are identified to control the five objectives of the walk: *cadence*, *step length*, *distance between feet*, *y offset* with respect to the baseline of walk, and *direction of walk* ( even if interacting each other, each of the five variables predominantly controls one of the five objectives). After each impact, at the start of step *k* they are

$$
\begin{aligned}
\delta_\omega(k) &\Rightarrow \omega(k) = \omega^+ + \delta_\omega(k)\,(cadence)\\
\delta_\alpha(k) &\Rightarrow \alpha(k) = \alpha_0 + \delta_\alpha(k)\,(step\ length)\\
\delta_{sway}(k)\cdot u + \delta_y(k) &\Rightarrow \alpha_z(k) = (\alpha_{z0} + \delta_{sway}(k))\cdot u + \delta_y(k)\\
&\quad (spacing\ between\ feet\ and\ y\ offset)\\
\delta_\gamma(k) &\Rightarrow \gamma(k) = \gamma^+ + \delta_\gamma(k)\,(direction)
\end{aligned}
\tag{30}
$$

where $u$ assume the values $+1, -1$ according to the right or left foot support.

It must be noted that no periodic reference is tracked. The whole gait style (cadence, length of the step, offset with respect to the baseline of walk-through a side shuffle, spacing between the two feet and direction) can be changed at each step. The energy consumption of the gait is measured by the difference, after the foot collision, of the kinetic energies before and after the application of the control (in particular $\delta_\omega(k)$ and $\delta_\gamma(k)$).

The next Figures 5 and 6 show a sample of a typical rectilinear walk, terminating with a halt. In particular, the Figure 7 show the energy needed at each step to maintain the gait, provided by $\delta_\omega(k)$ and how the gait collapses after two steps if no maintenance is performed.

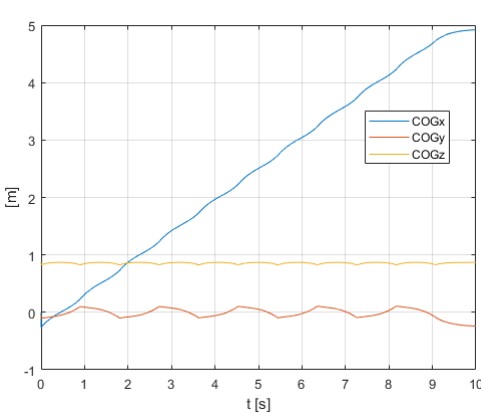

(**a**) COG along the x and y axes

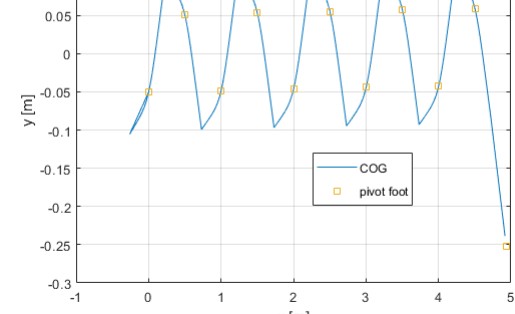

(**b**) Details of the projection of the COG on the ground and pivot foot posizion

**Figure 5.** The COG behaviour.

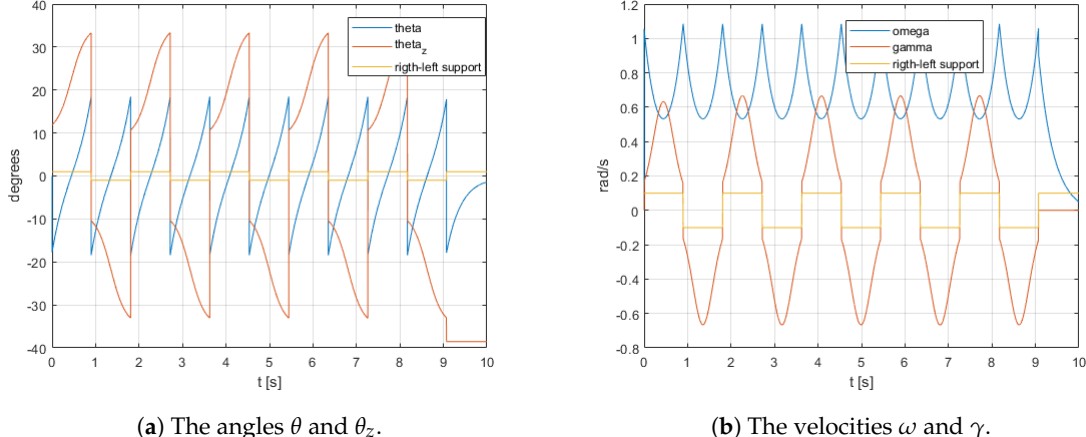

(**a**) The angles $\theta$ and $\theta_z$.

(**b**) The velocities $\omega$ and $\gamma$.

**Figure 6.** Angle position and velocity behaviors.

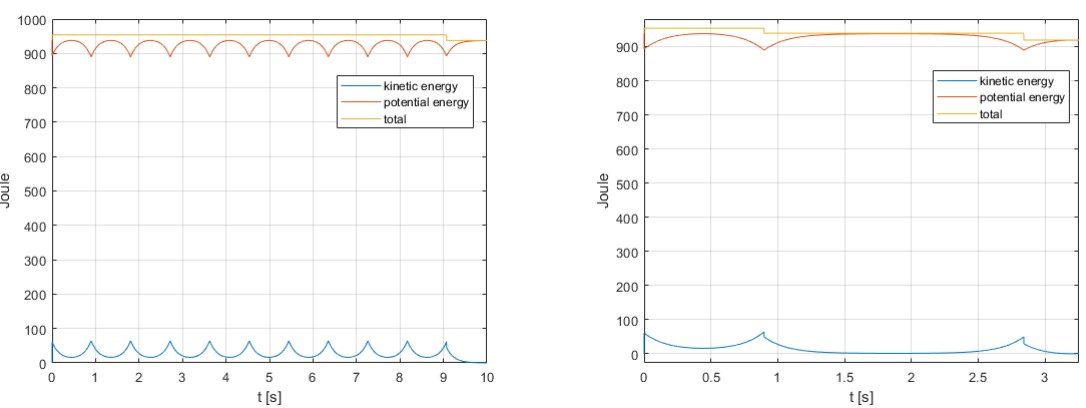

(**a**) The energies when the gait is maintained.

(**b**) The decay of the energies in absence of maintainment.

**Figure 7.** The energies.

### 7.1. Comparison between SIP and ZMP Based Gaits

An interesting question is how the SIP gait compares with the classical one based on the ZMP of a linear inverted pendulum. In previous papers, a 10/12 degrees of freedom biped model was simulated in rectilinear and curved trajectories [20]. The technique adopted was classical, by controlling the model to follow a preview trajectory based on the ZMP.

The total mass, the central inertia and the COG height of the biped were used to model the SIP. The parameters that control the gait of the SIP were adjusted to synchronize the two walks. To compare the center of pressure (COP) on the shoes in the two cases, the SIP is mounted on the ankle of the same foot of the biped (see Appendix A), no torque is transferred from the joints of the SIP, but the force on the ankle, returned from the non-holonomic constraint, are balanced by an identical force in the ZMP on the sole. The comparison is shown in the next figures.

In Figures 8a,b and 9 the two trajectories are superimposed. In particular, the right and left supports of the SIP and the periods of double support of the preview based simulation are also indicated.

The COG behaviours along the x a z axes are very similar, but not along the y axis. In fact, the details of the two COGs projected on the ground, and the relationships between the COP and the foot support placement, compared in the Figure 10a,b, show marked differences.

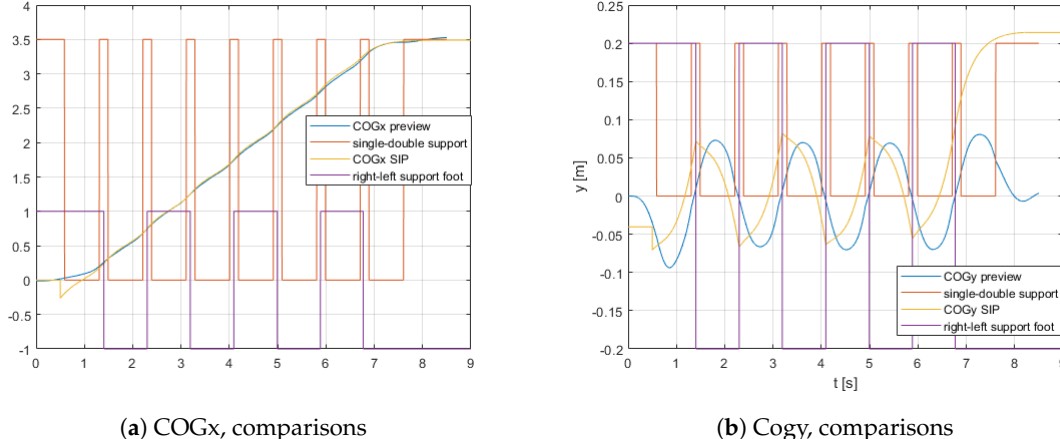

(**a**) COGx, comparisons

(**b**) Cogy, comparisons

**Figure 8.** Comparison of COGx and COGy.

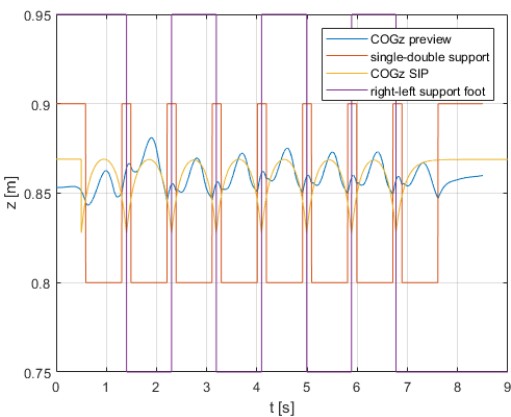

**Figure 9.** COGz, comparisons.

To achieve a similar sway of the *COG* on the *y* direction in the two cases, the SIP keeps the feet closer than in the preview case. Let consider that the SIP, passed the erect position is in free fall and the COP jumps suddenly on the new foot. Vice versa, in the preview-based gait the biped is always controlled to maintain the COP close to the supporting foot, and to transfer it, almost continuously from single to double support.

Finally, the energies were compared in Figure 11. As expected, the preview based on the ZMP demands a greater expenditure of energy.

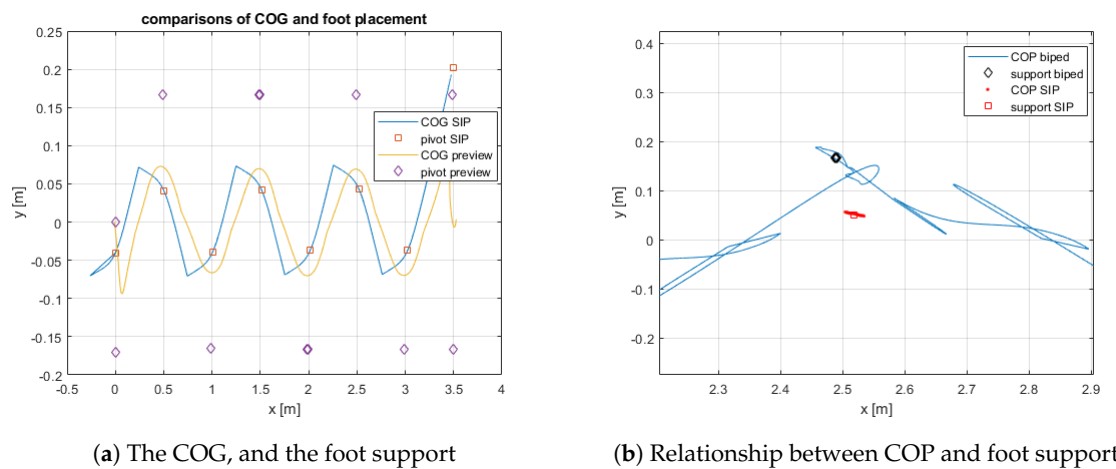

(**a**) The COG, and the foot support

(**b**) Relationship between COP and foot support

**Figure 10.** Comparison of COG and COP.

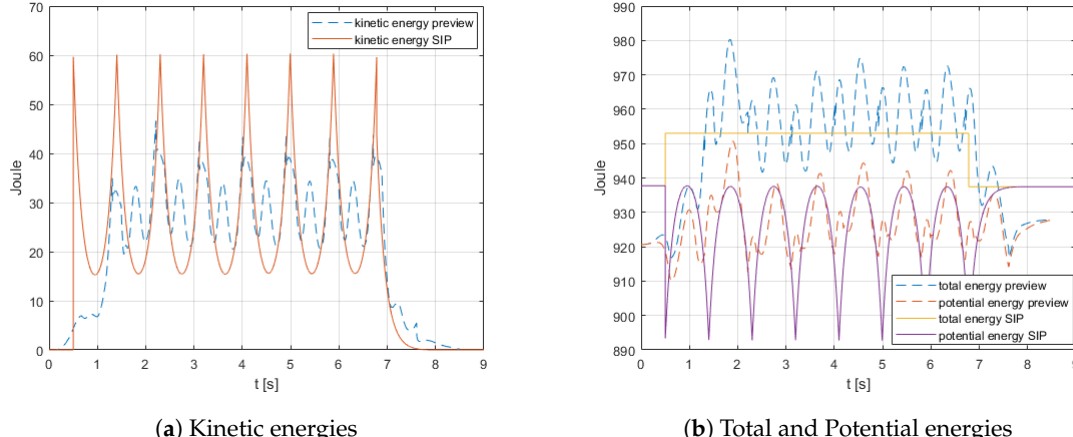

(**a**) Kinetic energies

(**b**) Total and Potential energies

**Figure 11.** Comparison of energies.

### 7.2. Turning While Walking

This last subsection shows how to generate turning while walking using the SIP model.

The walking trajectory is described in part III of [20]. It is obtained concatenating arcs of circle, whose ray can be changed at the beginning of each step. The control is based on a running local reference frame with respect to which the SIP is constrained to maintain a rectilinear walk. The local running frame, at each instant, rotates along the $z$ axis with respect to the world space to have its x axis tangent to the trajectory, and moves its origin to follow an involute of the path curve. Let $s(t)$ and $\dot{s}(t)$ be the curvilinear coordinate and its velocity on the path, their values also represent the motion on the abscissa of the local frame, and $\Theta_z(t)$ the orientation on the $z$ axis of the local frame at each instant $t$.

At time $T(k)$, immediately before the step $k$ starts, let indicate with $A_{xL}(k), A_{y_L}(k),$ $B_{xL}(k), B_{y_L}(k)$ the x and y coordinates of the last supporting foot, and of the swing foot at the contact (it is going to be the next support), in the local frame, respectively, and with $\theta_z(k) = \theta_z{}^+$ the current orientation angle after the contact.

$\delta_\omega$ is fixed to obtain the desired average *cadence* of the gait, $\delta_\alpha$ the average *step length*, and $\delta_{sway}$ the average *distance between feet*, averaging $1/(T(j) - T(j-1))$, $B_{xL}(j) - A_{xL}(j)$, $(B_{y_L}(j) - A_{y_L}(j)) \cdot u$, respectively, over the period $1 \leq j \leq k$. $(B_{y_L}(k) + A_{y_L}(k))/2$ measures the *offset* of the trajectory with respect to the centerline in the local frame and $(\theta_z(k-1) + \theta_z(k))/2$ the mean *direction* of walk at time $T(k)$.

From those measures a mild proportional, integral feedback is set to $\delta_y(k)$ and $\delta_\gamma(k)$ to maintain *offset* to zero, and *direction* to follow $\Theta_z(k)$. The result is the following Figure 12.

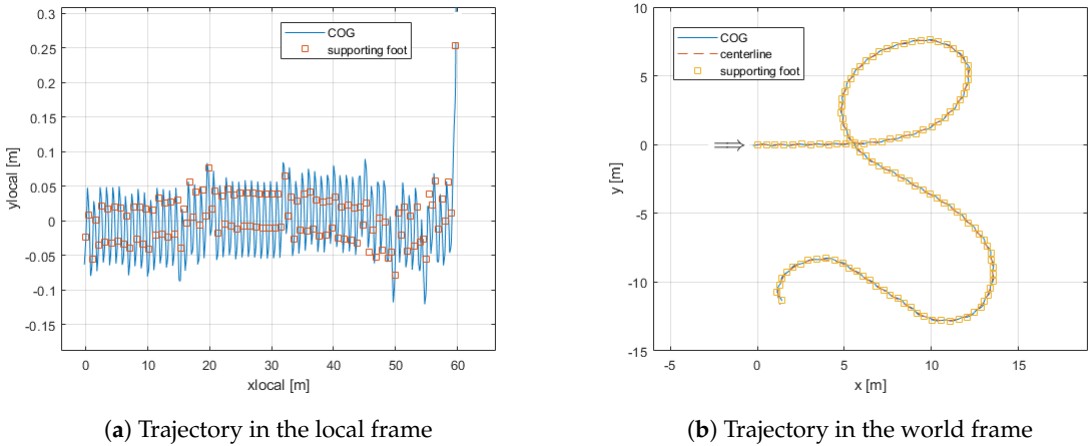

(**a**) Trajectory in the local frame

(**b**) Trajectory in the world frame

**Figure 12.** Trajectory-turn while walking.

## 8. Conclusions

Starting from the ideas of [4,5] of using a SIP model that emphasizes energies, the gait and the SFPE were reviewed with a solution based on the exact computation of the kinetic energy, the momentum projected on the z axis and the foot collision. Let note, that with this approach of SFPE the complicate handling of the projection of the angular momentum on the horizontal axis, contained in the original development, is not any more needed and arbitrary central inertia matrices are allowed. Consider that the solution of the impact (Equation (20)) offers the reaction forces at the collision, also. They are not used in this paper, but they will be in future extensions.

Very compact code for the equations have been obtained adopting the Kane's method and using his software environment. The solution of these equations, through a non-linear least square algorithm such as Levenberg-Marquardt, allows to introduce box bounds of the variables, to exploit the Jacobian, and to test for the feasibility of the solution. Moreover, the dependency of these equations on measurable angle positions and velocities and knowledge one step ahead of the ground level consents a real-time implementation to control robots or exoskeletons.

The introduction of the angle $\alpha_z$, taking into account the angle $\theta_z$, has extended the approach to non-periodic and omnidirectional gait and to recover the balance from a push in an arbitrary direction.

This paper, originally motivated by the need to help to recover balance, when disturbances are present, in a lower limb exoskeleton [26], is just the starting point. Future exentions, in order of complexity, are: introducing in the pantograph model the width of the pelvis in the distance between the two legs, adding a mass to the legs (sometime balance is recovered by inertia, just extending one leg), running and jumping.

The extension of the real-time control of the gait in the Cartesian space to all five control variables (30) with Model Predictive Control [27] will be also considered. So far we did not succeed, and only two variables have been controlled with classical PI techniques.

The approach was tested on a dynamic simulation of the SIP. Moreover, as the final objective is to offer a flexible Motion Generator for biped robot walking, the results have, also, been compared with the simulation of a fully actuated 12 DOF of a biped robot. In the comparison, the similarity of the COG behaviour in the sagittal plane suggests that actuated and underactuated controls can be intermixed [28], transforming the instantaneous increment of rotational velocities after the flying foot collision into a proper torque of the ankle of the supporting foot during double support and in the phase of single support, when the foot is flat. However, to obtain a nontrivial finite period of double support, compliance has to be added to the actual anelastic collision. As compliance is essential for that [11,29].

The behaviour of the COG in the frontal plane, that is consequence of a pure ballistic trajectory, will require a further investigation, as the resulting sway of the COG in the frontal plane seems excessive.

**Funding:** This research received no external funding.

**Conflicts of Interest:** The author declares no conflict of interest.

## Appendix A. A Kinetic Energy, Projection of the Angular Momentum on the Vertical Axis and Biped Parameters

The expressions of the kinetic energy and the angular moment on the vertical axis of the model in Figure 2a are the following

$$
\begin{aligned}
KE = {} & 0.5 \cdot (I22 + m \cdot L^2) \cdot \omega^2 \\
& + (I23 \cdot cos(\theta) - I12 \cdot sin(\theta)) \cdot \gamma \cdot \omega \\
& + 0.5 \cdot (I33 \cdot cos(\theta)^2 + I11 \cdot sin(\theta)^2) \cdot \gamma^2 \\
& - I31 \cdot sin(\theta) \cdot cos(\theta) \cdot \gamma^2 \\
& + 0.5 \cdot m \cdot L^2 \cdot sin(\theta)^2 \cdot \gamma^2
\end{aligned}
\tag{A1}
$$

$$k^\gamma = -2 \cdot I31 \cdot cos(\theta) \cdot sin(\theta) \cdot \gamma$$
$$(I23 \cdot cos(\theta) - I12 \cdot sin(\theta)) \cdot \omega$$
$$I33 \cdot cos(\theta)^2 \cdot \gamma \tag{A2}$$
$$I11 \cdot sin(\theta)^2 \cdot \gamma$$
$$m \cdot L^2 \cdot sin(\theta)^2 \cdot \gamma$$

The parameters of the model used in this paper, taken from [20], and representing a patient wearing an exoskeleton are shown in Figure A1.

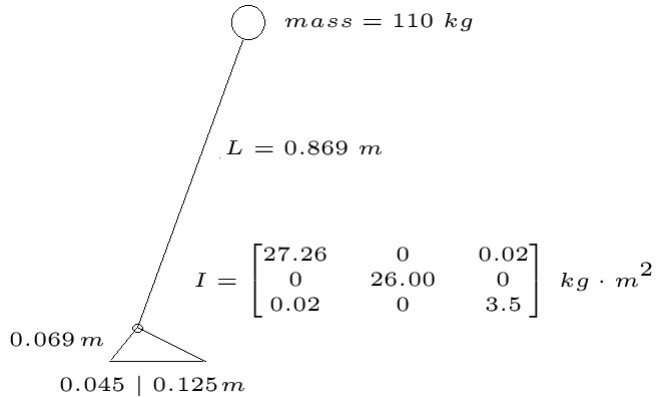

**Figure A1.** the biped parameters.

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
