# Peer review of "The Spherical Inverted Pendulum: Exact Solutions of Gait and Foot Placement Estimation Based on Symbolic Computation"

_applsci, doi:10.3390/app11041588_

Round 1
Reviewer 1 Report
ln 3. The approach is very attractive, however the authors introduced several questionable approximations.
What Authors? It may be worth identifying 2-3 key authors here so it appears less vague.
ln 4-6. Is the solution you are providing only viable for virtual environments? how does this model translate into real-world applications with variable conditions? Or, can it improve current methods in practical robots such as Atlas, ect.
ln 11. Provide a summary of your key findings at the end of your abstract, i.e. The results of this study indicate that....
Fig. 7-12. Probably best to centre these Figs as in Fig. 12 for consistency.
Your conclusion is rather short, mention the application of your solution i.e to recover balance (what is the significance of this?) and how does it broadens the current bottle-neck in bipedal robotics.
15 References is not many for a 17 page technical paper. I appreciate that most of the paper is taken up by your calculations in proving your solution. However, the paper would be significantly improved in your could provide more detailed/comparable studies in your opening sections for the sake of the reader.
Finally, there are a number of very long sentences i.e 17-19. It may be worth either condensing or rewriting these paragraphs for the ease of the reader.
Overall this is a solid and well structured piece of research well worth a read and of interest to engineers working in bipedal robotics.
Author Response
The whole paper has been revised for the style.
All the observations have been accounted for.
In particular three aspects have been considered:
- the introduction has been extended with a state of the art, and with the statement of the objectives of the paper;
- the conclusions indicate the future steps;
- the bibliography has been extended.
Reviewer 2 Report
The author has narrated the paper well with a defined research methodology. The paper has been framed well with a strong technical background. The overall presentation is average. However, the originality of the work is fine. Minor editing: Line 30: explicitely (spelling) Section2/3/4: heading starts with "the"(needs capitalization)Author Response
the whole paper has been revised for style.
Typos have been corrected
Reviewer 3 Report
The possibility of determining the reaction forces from (Eq20) is a great interest. In this regard, I think that article can be published for future researches.
Author Response
in relation with the impact several references and observation have been added in the development of the work and in the conclusions